# The Application of Magnetic Nanoparticles for Sentinel Lymph Node Detection in Clinically Node-Negative Breast Cancer Patients: A Systemic Review and Meta-Analysis

**DOI:** 10.3390/cancers14205034

**Published:** 2022-10-14

**Authors:** Pengcheng Liu, Jie Tan, Yuting Song, Kai Huang, Qingyi Zhang, Huiqi Xie

**Affiliations:** 1Laboratory of Stem Cell and Tissue Engineering, Orthopedic Research Institute, Med-X Center for Materials, State Key Laboratory of Biotherapy, West China Hospital, Sichuan University, Chengdu 610041, China; 2Department of Burn and Plastic Surgery, West China Hospital, Sichuan University, Chengdu 610041, China; 3Department of Orthopedics, West China Hospital, Sichuan University, Chengdu 610041, China

**Keywords:** breast cancer, sentinel lymph node, SPIO, nanomedicine

## Abstract

**Simple Summary:**

Generally, the standard method of applying sentinel lymph node biopsy in breast cancer patients is via a technetium-labeled nanocolloid (radiolabeled tracer) with or without blue dye. However, the radioactive agents may cause challenges both to hospitals and patients. Alternatively, a safer candidate method, such as SPIO, has been often introduced and validated in comparison with standard methods. The aim of this study was to perform a systematic review and meta-analysis to evaluate the diagnostic accuracy of SPIO and its clinical impact on the management of breast cancer. Based on our study, SPIO could be considered as an alternative standard of care for sentinel lymph node detection. Compared with the standard method, SPIO exhibited equivalent or even superior detection capacities, while safety could be guaranteed. Lower SPIO doses would also not impair detection capacity compared with the standard method. The absence of radioactivity of SPIO is one of the most important advantages for clinical applications.

**Abstract:**

Superparamagnetic iron oxide (SPIO), an alternative mapping agent, can be used to identify sentinel lymph nodes in patients with clinically node-negative breast cancer. However, its performance in comparison with the standard method, using a radioisotope (technetium-99 m, Tc) alone or in combination with blue dye, remains controversial. Hence, a systematic review and meta-analysis were conducted to evaluate the diagnostic accuracy of SPIO and its clinical impact in the management of breast cancer. The PubMed, Embase, and Cochrane databases were comprehensively searched from inception to 1 May 2022. Cohort studies regarding the comparison of SPIO with standard methods for sentinel lymph node identification were included. A total of 19 prospective cohort studies, which collectively included 2298 clinically node-negative breast cancer patients undergoing sentinel lymph node identification through both the standard method and SPIO, were identified. The detection rate for sentinel lymph nodes (RR, 1.06; 95% CI, 1.05–1.08; *p* < 0.001) was considerably higher in the SPIO cohorts than in the standard method cohorts, although this difference was not significant in detected patients, patients with positive sentinel lymph nodes, or positive sentinel lymph nodes. Compared with the standard method, the SPIO method could be considered as an alternative standard of care for sentinel lymph node detection in patients with clinically node-negative breast cancer.

## 1. Introduction

Sentinel lymph node biopsy (SLNB) is routinely used to confirm the metastatic status of the axilla in patients who are clinically presented with a node-negative disease in most early-stage breast cancers. Generally, the standard method of applying SLNB is via subareolar, periareolar, or peritumoral injection of a technetium-labeled nanocolloid (radiolabeled tracer, RI) with or without blue dye (BD) [1]. Currently, a plethora of trials have demonstrated the feasibility and accuracy of this available method [2,3], some even with many benefits in patients simultaneously [4,5,6].

Nevertheless, the radioisotope in these agents may bear logistical challenges to hospitals, including the handling and disposal of isotopes, as well as training staff and legislative permission burdens [7]. The relatively short half-life (about 6 h) of the isotope restricts theatre scheduling since the injection is usually performed by nuclear medicine staff, not by surgeons themselves. Patients may express reluctance to radiation exposure, especially in pregnancy for fear of fetal radiation exposure. In addition, the complication of anaphylaxis, the risk of obscuring the surgical field, and the potential risk of teratogenicity also limits the usage of BD under such circumstances [8,9]. Therefore, alternative safer candidate methods are extremely limited and desperately needed. For that reason, several techniques are currently being tried and evaluated [10,11].

Among those techniques, superparamagnetic iron oxide (SPIO) emerges as a kind of magnetic nanoparticle that is made from superparamagnetic iron oxide coated with carboxydextran [12]. Clinically, it is often administrated with saline before or after anesthesia induction. After injection, these nanoparticles can be further taken up by the lymphatics traveling to the lymph nodes, where a handheld magnetometer can be utilized to detect their occurrence [13]. In this way, the identification of the SLNs could be confirmed by a handheld magnetometer and the color change (brownish or black) where the nodes are stained can be visualized [7,14].

The advantages of SPIO include a long shelf life and nonradioactive properties without requiring special handling procedures, and thus it can be administered by surgeons with easy access. Furthermore, problems regarding waste disposal are also solved [15]. Up to now, several trials have evaluated the outcomes of applying this technique to complement and optimize the SLNB strategy in practice [16]. However, the definitive conclusion remains controversial. Therefore, we comprehensively and rigorously performed this systematic review and meta-analysis, aiming to figure out whether the performance and utility of the magnetic nanoparticle technique may be a substitute for the standard technique.

## 2. Materials and Methods

We conducted a meta-analysis of cohort studies to assess whether SPIO is a potential substitute for the standard method for sentinel lymph node (SLN) detection in patients with clinically node-negative breast cancer. This study followed the Preferred Reporting Items for Systematic Reviews and Meta-Analyses (PRISMA) reporting guideline [17]; the protocol was registered on PROSPERO (CRD: 42020218583).

The PubMed, Embase, Web of Science, and Cochrane databases were searched from inception to 1 May 2022. A combination of Medical Subject Headings and free-text terms, including their variants, was searched in those databases. The Medical Subject Heading (MeSH) terms used were (breast cancer’ AND ‘sentinel lymph node’ AND (‘nanoparticle*’ OR ‘magnet*’ OR ‘magnetic*’ OR ‘sentimag’)). Search strategies were adapted to each database’s search engine and were limited to those in the English language and involving human participants. All articles were reviewed based on their titles and abstracts, and those meeting the inclusion criteria were then read in full text. Complementally, the references of the included articles were also searched manually for new candidate articles. The detailed search strategies are shown in the Appendix A.

Studies were included if they fulfilled the following inclusion criteria: prospectively conducted clinical trials in which the efficacies between magnetic technique and standard technique (using either radiolabeled tracer alone or a combination of BD) for SLNB were compared in patients with breast cancer; ethical approval obtained; sample size greater than 10. Articles without available full text were excluded, as were conference abstracts, review articles, case reports, editorial reports, and letters to the editor.

Each study was evaluated for eligibility by two reviewers (L.PC. and S.YT.), and study characteristics and outcomes for all selected studies were extracted. A third reviewer (T.J.) meticulously verified the accuracy of the extracted data. The items of these studies that might potentially be related to the outcomes were extracted as follows: first author, publication year, participant characteristics, injection time and dose of SPIO, number of detected patients and positive patients (patients with metastatic sentinel lymph nodes), number of detected sentinel lymph nodes and positive sentinel lymph nodes (metastatic sentinel lymph nodes), as well as other baseline characteristics. When trials had a multifactorial design comparing multiple invention groups, we extracted data and assigned them to the relevant intervention group. We treated each intervention group independently within our analysis [18,19].

All extracted data were tabulated and presented as means and percentages. The R package (version 4.1.3) was used for the quantitative analyses. For (positive) SLN identification, the Cochran–Mantel–Haenszel method was used to test for an association between the two techniques. A similar method was used for detected patients and patients with positive SLNs, across all studies.

We assessed heterogeneity between studies statistically with the I² statistic for inconsistency. We used forest plots to estimate the statistical heterogeneity among studies and subgroup analyses. We used I² levels of 25%, 50%, and 75% to represent low, moderate, and high heterogeneity [20]. A fixed- or random-effects model was used to account for pooled relative risk (RR) according to their respective heterogeneity (I^2^ < 50%, fixed-effects models; I^2^ > 50%, random-effects models). The probability for each variable was computed and the RR with a 95 percent confidence interval (CI) was calculated for binary data variables. We considered the comparison significant if *p* < 0.05 (all tests were two-tailed).

Subgroup analysis, based on the injection dose of the magnetic tracer and the selection of the standard method (RI + BD, RI, or RI ± BD), was also performed to analyze the related efficacy. With a similar initial concentration (Appendix A), the dosage-related subgroup analyses were stratified through baseline injection volumes of 0.5 mL, 1.0 mL, 1.5 mL, or 2.0 mL. Of the 19 studies, 1 study was excluded from the subgroup analysis of doses because it randomized participants into three groups according to the usage of doses [21].

The risk of bias in these included non-randomized studies of interventions was assessed using the Risk Of Bias In Non-randomized Studies of Interventions (ROBINS-I) [22]. Seven ‘signaling questions’ help users judge the risk of bias within each domain. The judgments within each domain carry forward to an overall result across bias domains for the outcome being assessed. Two reviewers (L.PC. and S.YT.) performed the assessments independently. In the case of a disagreement, a consensual decision was counseled by the third author (T.J.).

## 3. Results

### 3.1. Description of Studies

The flow of the selection process is shown in Figure 1. Our meta-analysis retrieved 436 articles. After removing 159 duplicates, screening of the 277 titles and abstracts was performed and a total of 235 articles were excluded; only 42 articles remained. Further, nine articles were excluded owing to the invalidity of the reported outcomes for not meeting the inclusion criteria. Moreover, nine articles without parallel controls, four articles compared in two independent cohorts, and two articles compared with the non-standard method were also excluded after the full text assessment. Ultimately, 19 studies from 18 articles were incorporated into the meta-analysis.

### 3.2. Quality Assessment

The quality of the 19 trials was assessed using the Risk Of Bias In Non-randomized Studies of Interventions (ROBINS-I). All of the studies (Appendix A) were graded as at low risk of bias in the classification of interventions, bias due to deviation from intended interventions, bias due to missing data, and bias in selection of the reported result. Three studies were considered at moderate risk of bias in the measurement of outcomes, one study was considered at moderate risk of bias due to confounding, and one study was considered at moderate risk of bias due to selection of the reported result. Overall, 5 of 19 were ranked as at moderate risk of bias, while the other 14 studies were rated as low risk.

The main characteristics and specific interventions of the included studies are presented in Table 1. In summary, 19 studies from the included 18 articles reported a total of 2298 patients with clinically node-negative breast cancer who received SLNB procedures, with their pooled detection rate at 97.2% (2234/2298) in SPIO cohorts and 96.9% (2226/2298) in standard cohorts. A total of 15 studies reported 488 patients with positive SLNs in a total of 1966 patients, where procedures succeeded in 96.3% (470/488) of SPIO cohorts and 95.5% (466/488) of standard cohorts. 

An amount of 2298 patients from the 19 studies harvested 4367 nodes (1.9 nodes per patient) in total, and the retrieved rates were 90.9% (3971/4367) via SPIO and 85.7% (3741/4367) via the standard method for the SLN detection. Pathological metastasis was confirmed in 639 of 2861 SLNs (22.3%), with their corresponding rates being 96.7% (618/639) and 93.9% (600/639) for these positive SLNs detected via SPIO and the standard method (listed in Table 2).

#### Sentinel Lymph Node Biopsy Procedure

The generally administered doses of SPIO were 0.5 mL [24,32], 1.0 mL [31,34], 1.5 mL [34], and 2.0 mL [7,12,14,21,23,25,26,27,28,29,30,33,35,36] for clinical usage, usually diluted with saline and administered through subcutaneous injection into the peritumoral [32], intratumoral [24], or periareolar or subareolar zones [7,12,14,21,23,25,26,27,28,29,30,31,32,33,34,35,36]. The optimal time for administration of this agent was at least 20 min ahead of the axillary surgery [7,12,14,21,23,25,27,29,30,32,33,34,35,36], but this time could be extended in advance to several hours [24,28,31] or even some days [26,34] before the surgical procedure was taken. After the completion of this injection, an additional 5 min massage of the injected area was suggested for promoting the distribution of magnetic tracers. For the standard studies, a radiolabeled tracer (with or without BD) was administered in accordance with each center’s protocol, with no technique or dose difference in the radioisotope between centers. Among these, 13 studies were additionally administered with BD [7,12,23,24,25,26,27,28,29,30,33,34,36], and the details of the tracer and method of injection were provided in all studies (Table 2).

During surgery, the surgeon used the handheld magnetometer for skin localization of the sentinel lymph node, which was followed by a gamma probe to confirm the position. Among them, thirteen studies used the magnetometer with a combination of standard techniques for in vivo confirmation [14,21,25,27,28,29,30,32,33,34,35,36]. Six studies used the standard technique only after SLNB was performed with the magnetometer, as well as for ex vivo verification. Six studies had performed lymphoscintigraphy before SLNB [14,21,23,28,29,33]. Eleven studies did not report whether lymphoscintigraphy had been taken [7,12,21,24,25,30,31,32,34,35,36], while the remaining studies did not take lymphoscintigraphy routinely [26,27]. Of note, two studies which blinded the surgeons when executing the lymphoscintigraphy were later resolved by the introduction of the magnetic technique [23,28].

### 3.3. Identification and Node Retrieval

#### 3.3.1. Detection Rate for Patient

A total of 19 studies were included when analyzing the outcome of the patient detection rate [7,12,14,21,23,24,25,26,27,28,29,30,31,32,33,34,35,36]. After our pooled meta-analysis with a fixed-effects model, no significant differences were revealed in patient detection rates between the SPIO and standard method group (RR, 1.00; 95% CI, 0.99–1.01; I^2^ = 20%; *p* = 0.21) (Figure 2).

No comparable differences in patients’ detection rate between the SPIO and standard method groups were detected in women with respective injection volumes of 0.5 mL (2 studies, RR, 0.99; 95% CI, 0.93–1.06; I2 = 79%; *p* = 0.03), 1.0 mL (2 studies, RR, 0.98; 95% CI, 0.96–1.00; I2 = 67%; *p* = 0.08), 1.5 mL (1 study, RR, 1.00; 95% CI, 0.99–1.01; I2 = 24%; *p* = 0.17), or 2.0 mL (13 studies, RR, 1.02; 95% CI, 1.00–1.02; I2 = 5%; *p* = 0.40) (Figure 3). In addition, the selection of the standard method did not influence outcomes (Appendix A).

#### 3.3.2. Detection Rates for SLNs

In this section, 17 studies were included [7,12,21,23,24,26,27,28,29,30,31,32,33,34,35,36]. After the pooled meta-analysis, the means of the retrieved sentinel lymph nodes (SLNs) were 1.9 and 2.0 per patient in the standard method group and SPIO, respectively. The studies showed high heterogeneity (I2 = 85% percent); hence, a random-effects model was used. Compared to the standard method, the SPIO group significantly showed its superiority over the standard group in harvesting more SLNs (RR, 1.06; 95% CI, 1.05–1.08; *p* < 0.001) (Figure 4).

The SPIO group noticeably outperformed the standard method in detection rates for SLNs with respective to injection volumes of 1.0 mL (1 study, RR, 1.16; 95% CI, 1.11–1.22; *p* < 0.001) or 2.0 mL (12 studies, RR, 1.06; 95% CI, 1.03–1.08; I^2^ = 65%; *p* < 0.001), while no comparable difference was noticed in the dose of 0.5 mL (2 studies, RR, 1.11; 95% CI, 0.65–1.90; I^2^ = 97%; *p* < 0.001) or 1.5 mL (1 study, RR, 0.98; 95% CI, 0.92–1.04; *p* < 0.001) (Figure 5).

The SPIO group was superior to RI (six studies, RR, 1.16; 95% CI, 1.03–1.31; I2 = 90%; *p* < 0.001) in detection rates for SLNs, while the differences were unremarkable in comparison with RI + BD (eight studies, RR, 1.02; 95% CI, 0.96–1.07; I2 = 80%; *p* < 0.001) and RI ± BD (three studies, RR, 1.06; 95% CI, 1.03–1.09; I2 = 0%; *p* = 0.42) (Appendix A).

##### Detection Rates for Patients with Positive SLNs

This comparison incorporated 15 studies [7,12,14,21,23,24,26,27,28,29,30,34,35,36]. After our pooled meta-analysis with a fixed-effects model, the results showed no significant difference in detecting patients with positive SLNs between the SPIO and standard method (RR, 1.01; 95% CI, 0.98–1.04; I^2^ = 0%; *p* = 0.85) (Figure 6).

##### Detection Rates for Positive SLNs

In this section, 10 studies were incorporated for final analysis [12,14,27,28,29,30,32,34,35,36]. After analysis with a fixed-effects model, SPIO was noninferior to the standard method in detection rates for positive SLNs (RR, 1.03; 95% CI, 1.00–1.06; I^2^ = 29%; *p* = 0.18) (Figure 7).

##### Complications

SLNB-related adverse events were recorded in 16 studies [7,14,21,23,24,25,26,27,28,29,30,31,32,34,36]. A brownish or grayish coloration of the breast skin was recorded post-operation, with a complication rate ranging from 16.3% to 84.4% in a total of 12 studies [7,21,23,24,25,27,28,29,30,31,34,36]. Of note, one study did not report skin pigmentation after the operation because the injected site was resected during surgery [32], while the other six remaining studies did not record any dye- or tracer-related skin-staining events. However, within a period of follow-up (usually 1 to 15 months) most of the skin pigmentation would be attenuated or have vanished, but a small portion of patients would remain unchanged or even be slightly enlarged [23,27,29,31,34]. No allergic or systemic inflammatory reactions were documented in all studies.

Disadvantages of the magnetic technique were also discussed in some studies [7,12,14,23,27,28,29,32,34,36]. For instance, the relatively large diameter of the magnetometer’s handheld probe would result in larger surgical incisions; the time-consuming frequent balancing of the magnetic baseline level required a correct localization; detection of the local lesion at revision surgery would be challenging for surgeons unfamiliar with the technique, particularly in patients with mastectomies [30]. Moreover, the requirement for the use of plastic alternatives instead of standard surgical retractors, the role of lymphoscintigraphy in successful SLN localization, and the risk of potentially hindering the diagnostic performance of follow-up breast magnetic resonance imaging (MRI) were also reported to be their non-negligible shortcomings [14,21].

## 4. Discussion

In this systematic review and meta-analysis, a total of 19 studies on the utilization of SPIO in clinically node-negative breast cancer was included and finally confirmed that SPIO could be an excellent substitution for the standard method whether for identification rates or process improvement. At the same time, we proved lower doses of SPIO have comparable diagnostic accuracy to the standard method. Interestingly, in trials where discordance in detection existed, a greater number of SLNs were identified via the SPIO group compared to the standard group. This significant revelation needs to be noticed because the number of SLNs identified has important implications for the accuracy of the ongoing procedure.

It is noted that successful identification has been influenced by many factors. Different injection doses, sites (peritumoral, intratumoral, periareolar, or subareolar), and timeframes (perioperative or preoperative) might have an impact on the lymphatic uptake of the magnetic tracer. Previous studies reported different injection doses (0.5 mL [24,32], 1.0 mL [21,31,34], 1.5 mL [21,34], and 2.0 mL [7,12,14,21,23,25,26,27,28,29,30,33,35,36] of injection volume) and injection sites (peritumoral [32], intratumoral [24], or periareolar or subareolar [7,12,14,21,23,25,26,27,28,29,30,31,32,33,34,35,36]) have noninferior detection efficiency compared with the standard method. The SLN detection rate per patient was constantly comparable to the standard method, which was also unaffected by SPIO dose [21,30]; a 0.5 mL volume of SPIO was sufficient for SLN identification [37]. This finding is consistent with the current subgroup analysis, nevertheless, high doses (2.0 mL of injection volume) of SPIO still reported a superior performance compared to the standard method in detecting more SLNs (Figure 5).

Accordingly, the relationship between the time point choice of administration of the SPIO and the final detection rate has also been explored. Perioperatively [7,12,14,21,23,24,25,27,28,29,30,31,32,33,34,35,36], one or more days preoperative [26,31,34] injection was comparable to the standard technique in the detection rate. Additionally, higher detection rates than the traditional technique were found if SPIO was injected 1–28 days ahead of surgery compared to that administered on the day of surgery [26,38]. SPIO showed no significant difference in the detection rates between subareolar and peritumoral injections [39]. In another study, a lower SPIO volume injected up to 7 days before the operation has comparable efficacy to the higher SPIO dose (2.0 mL) group and the standard method group for SLN detection [34]. A peritumoral injection and a smaller SPIO dose might also be helpful for addressing the concern of postoperative MRI artifacts [40]. The present results provide convincing evidence that not only a lowered dose but also a flexible injection timeframe in the preoperative period might be combinatorially adopted to enhance the detection rate and SLN retrieval.

SPIO particles do not require any special storage and there are no radiation exposure risks, neither for healthcare personnel nor for the patient. Therefore, non-metal instruments might be adopted to prevent interference between the metal and detection probe. This may present a challenge in obese patients because additional instruments are required to facilitate the dissection of the deep axilla. In addition, a heavier cost burden owing to the equipment of nonmagnetic surgical instruments is also one of their drawbacks, but this could be offset by expense reduction from streamlining processes. Additionally, sparing sentinel lymph node dissection (SLND) procedures were applicable in women with a susceptibility axillary status [25,36,38]. In addition, preoperative lymphoscintigraphy before SLNB was inconsistently used [1,2]. An equally effective outcome was observed in a recent prospective, multicenter, randomized phase III trial, followed by a simplified preoperative workflow and reduced cost, irrespective of the preoperative lymphoscintigraphy results [41].

Another concern is that the depth that the magnetometer can reach is noncomparable to that of the gamma probe, possibly due to the disparate probe size and differential capacity, which can influence the identification of the deeper nodes [24,42]. To this end, novel magnetometry has advanced the appropriate size, excellent detection sensitivity, and good attenuation of noise, as well as the high spatial sensitivity (sufficient depth resolution) to fulfill the pressing clinical requirement [43,44,45]. Moreover, owing to the interference, magnetic nanoparticles should not be used in patients who need MRI to diagnose occult lesions, assess treatment response, or undergo surveillance [40,46]. Of note, the remnants of such nanoparticles would diminish over time and the image quality could be further improved by modifying the sequences [47]. Furthermore, the impact of the differently administered dose of SPIO on skin staining and MRI artifacts is currently studied as well (ISRCTN85167182).

Safety issues regarding the application of SPIO must also be carefully considered. No severe allergic reactions were reported in any of the currently published trials; this can partly be explained by the exclusion of patients with hypersensitivity to iron or dextran compounds and those with pacemakers or metal implants. Dermopigmentation is the most frequently reported complication, with a rate of up to 20% in the injection site, similar to what is found after the injection of blue dye [23,28,48,49]. Skin staining after SPIO injection is a concern as well, even though several reports showed that most patients did not consider it an upsetting problem [21,25]. As reported elsewhere, a deeper injection would be instrumental in reducing the incidence of discoloration [29]. In addition, in a large-scale study, no toxicity in radiotherapy or chemotherapy after SLNB via SPIO was observed [50]. Even though encouraging results have been observed regarding the feasibility of the magnetic technique, further studies with a larger sample size, prospective nature design, as well as rigorous methods of outcome ascertainment, should be warranted before wide clinical implementation [16].

Some limitations also inevitably existed, such as variable blue dyes in the standard method and different cut-off points in applying the SPIO signal to detect the SLNs. However, as clinicopathologic reports were not consistently reported across the studies, these parameters were not included or incorporated in the final comparison. Further, the patient-reported outcomes measures (PROMs), an integral component of benefit–risk assessments in the evaluation of new treatment regimens, were generally neglected in most of the studies; this may miss some information and cause some biases in drawing the conclusions. Therefore, randomized trials with a larger sample size comparing the SPIO technique to the standard tracers on locoregional recurrence and survival benefit are desperately needed in the future to advance this technique to benefit the patients who really need it.

## 5. Conclusions

Overall, by pooling the trial data, this meta-analysis provides increased evidence that SPIO could be an alternative method to the standard method for axillary node mapping in breast cancer and suggests generalizability of the technique to a wider population.

## Figures and Tables

**Figure 1 cancers-14-05034-f001:**
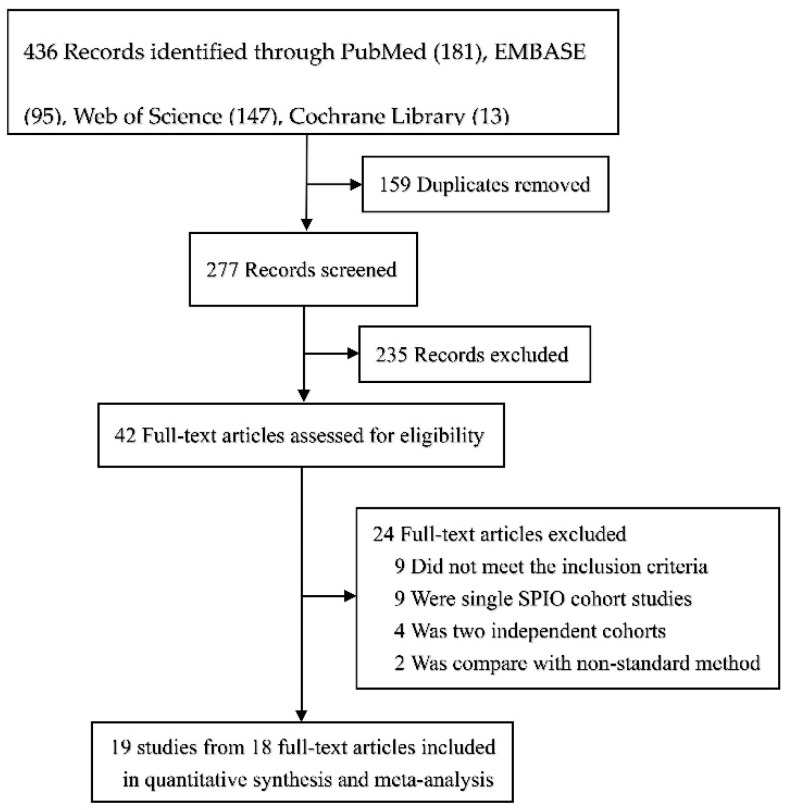
Flowchart depicting the selection process for the inclusion of studies in the article.

**Figure 2 cancers-14-05034-f002:**
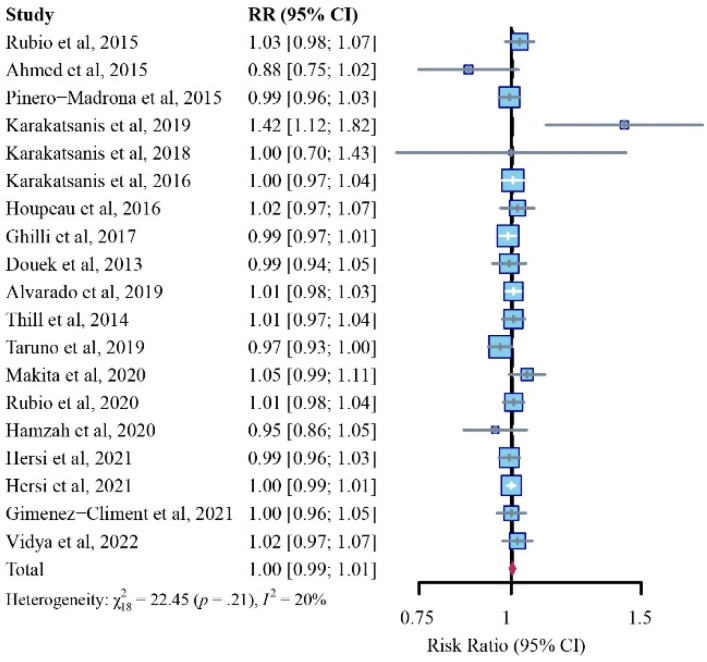
Relative risk (RR) analysis of patient identification rates for SPIO versus standard method (fixed-effects model).

**Figure 3 cancers-14-05034-f003:**
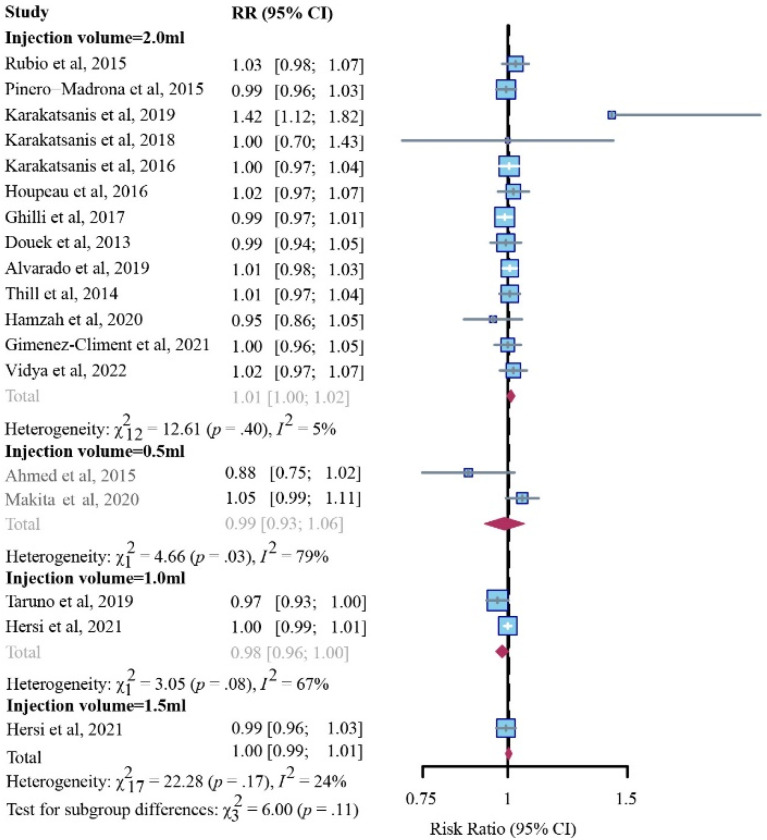
Subgroup analysis of patient identification rates for SPIO versus standard method (fixed-effects model), stratified through baseline injection volumes of 0.5 mL, 1.0 mL, 1.5 mL, or 2.0 mL.

**Figure 4 cancers-14-05034-f004:**
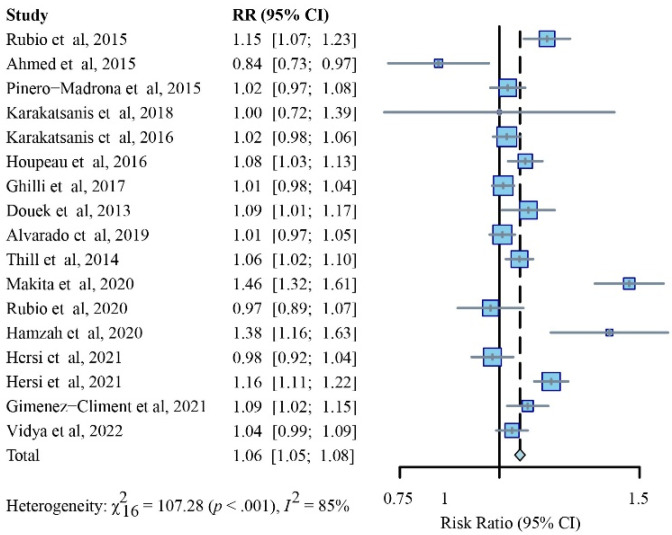
Relative risk (RR) analysis of SLN identification rates for SPIO versus standard method (random-effects model).

**Figure 5 cancers-14-05034-f005:**
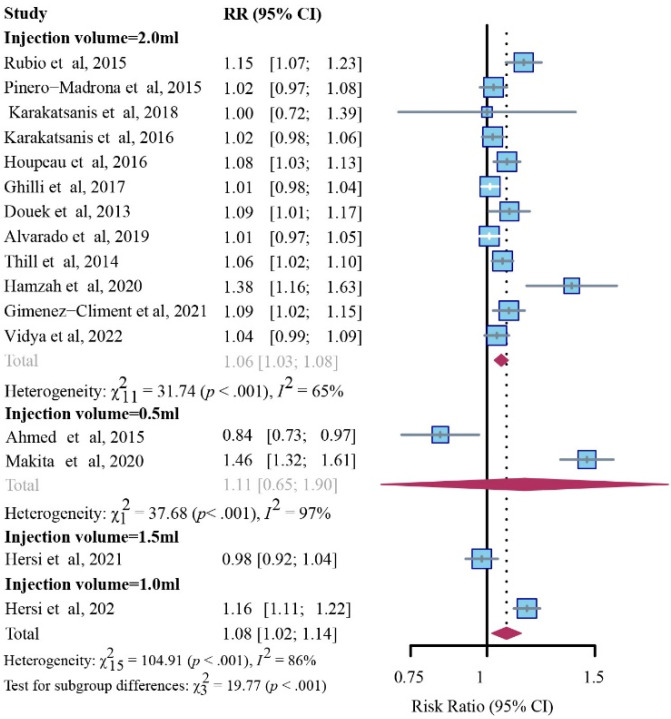
Subgroup analysis of SLN identification rates for SPIO versus standard method (random-effects model), stratified through baseline injection doses.

**Figure 6 cancers-14-05034-f006:**
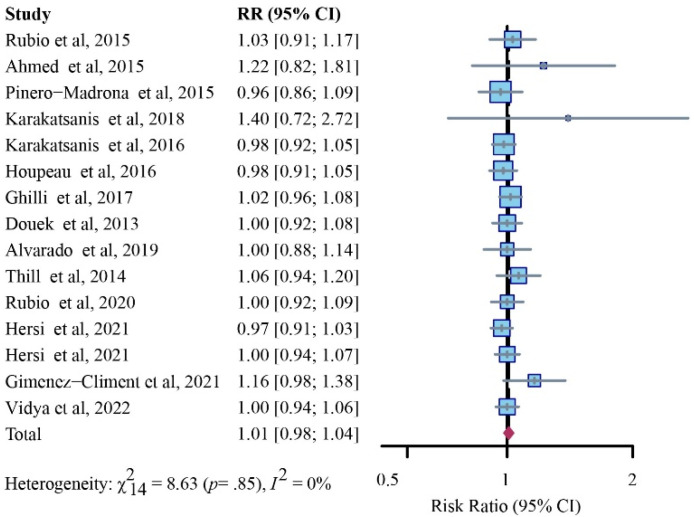
Relative risk (RR) analysis of identification rates for patients with positive SLNs via SPIO versus standard method (fixed-effects model).

**Figure 7 cancers-14-05034-f007:**
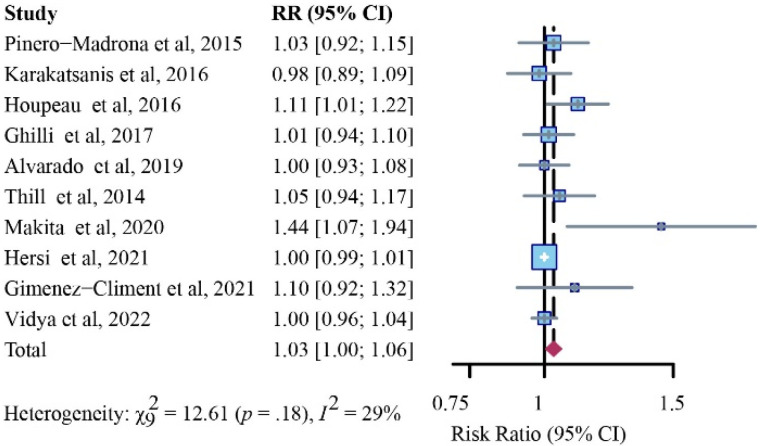
Relative risk (RR) analysis of positive SLN identification rates for SPIO versus standard method (fixed-effects model).

**Table 1 cancers-14-05034-t001:** Baseline characteristics.

Author and Year	Country	Methods Compared	Injection Site	Injection Volume	Interval Time *
Rubio et al., 2015 [23]	Spain	SPIO, RI	subareolar	2.0 mL	20–25 min
Ahmed et al., 2015 [24]	UK	SPIO, RI + BD	intratumoral	0.5 mL	Within 24 h
Pinero−Madrona et al., 2015 [12]	Spain	SPIO, RI + BD	peritumoral or periareolar	2.0 mL	10 to 85 min
Karakatsanis et al., 2019 [25]	Sweden	SPIO, RI + BD	subareolar	2.0 mL	At least 20 min
Karakatsanis et al., 2018 [26]	Sweden	SPIO, RI + BD	subareolar	2.0 mL	3–15 days (median 8 days)
Karakatsanis et al., 2016 [27]	Sweden	SPIO, RI + BD	subareolar	2.0 mL	At least 20 min
Houpeau et al., 2016 [28]	France	SPIO, RI ± BD	periareolar	2.0 mL	At least 20 min
Ghilli et al., 2017 [29]	Italy	SPIO, RI + BD	subareolar	2.0 mL	At least 20 min
Douek et al., 2013 [7]	UK and Netherlands	SPIO, RI ± BD	periareolar	2.0 mL	At least 20 min
Alvarado et al., 2019 [30]	America	SPIO, RI + BD	subareolar	2.0 mL	At least 20 min
Thill et al., 2014 [14]	Germany	SPIO, RI	periareolar or peritumorally	2.0 mL	At least 20 min
Taruno et al., 2019 [31]	Japan	SPIO, RI	subareolar	1.0 mL	1 day
Makita et al., 2020 [32]	Japan	SPIO, RI	periareolar or peritumorally	0.5 mL	At least 20 min
Rubio et al., 2020 [21]	Spain	SPIO, RI	subareolar	1.0 mL, 1.5 mL, or 2 mL	37 min (range 35–39 min)
Hamzah et al., 2020 [33]	Singapore	SPIO, RI + BD	subareolar	2.0 mL	At least 20 min
Hersi et al., 2021 [34]	Sweden	SPIO, RI + BD	periareolar or peritumorally	1.5 mL	At least 20 min
Hersi et al., 2021 [34]	Sweden	SPIO, RI + BD	Periareolar or peritumorally	1.0 mL	1–7 days
Giménez-Climent et al., 2021 [35]	Spain	SPIO, RI	subareolar	2 mL	At least 20 min
Vidya et al., 2022 [36]	UK	SPIO, RI ± BD	subareolar	2 mL	At least 20 min

* Time intervals from SPIO injection to axillary surgery. Abbreviations: SPIO, superparamagnetic iron oxide; RI, radioactive isotope; NR, not reported; min, minutes.

**Table 2 cancers-14-05034-t002:** Main outcomes of the incorporated articles.

Author and Year	Patients			SLNs			Positive Patients			Positive SLNs		
	Total	SM	SPIO	Total	SM	SPIO	Total	SM	SPIO	Total	SM	SPIO
Rubio et al., 2015 [23]	118	95.7%	98.3%	287	80.1%	92.0%	36	91.7%	94.4%	NR	NR	NR
Ahmed et al., 2015 [24]	33	97.0%	84.8%	67	92.5%	77.6%	5	80.0%	100.0%	NR	NR	NR
Pinero-Madrona et al., 2015 [12]	181	97.8%	97.2%	321	89.3%	92.5%	60	91.7%	88.3%	76	88.2%	90.8%
Karakatsanis et al., 2019 [25]	40	65.0%	92.5%	NR	NR	NR	NR	NR	NR	NR	NR	NR
Karakatsanis et al., 2018 [26]	12	83.3%	83.3%	16	81.3%	81.3%	3	66.7%	100.0%	NR	NR	NR
Karakatsanis et al., 2016 [27]	206	97.1%	97.6%	402	91.3%	93.3%	54	99.4%	96.3%	68	92.6%	91.2%
Houpeau et al., 2016 [28]	108	95.4%	97.2%	220	90.2%	97.2%	46	97.8%	95.7%	61	88.5%	98.4%
Ghilli et al., 2017 [29]	193	99.0%	97.9%	308	94.7%	95.8%	57	96.5%	98.3%	77	93.5%	94.8%
Douek et al., 2013 [7]	160	95.0%	94.4%	404	73.5%	80.0%	35	97.1%	97.1%	NR	NR	NR
Alvarado et al., 2019 [30]	146	98.6%	99.3%	369	93.5%	94.3%	22	95.5%	95.6%	25	100.0%	100.0%
Thill et al., 2014 [14]	150	97.3%	98.0%	291	91.8%	97.3%	42	91.2%	97.1%	45	91.1%	96.6%
Taruno et al., 2019 [31]	210	98.1%	94.8%	NR	NR	NR	NR	NR	NR	NR	NR	NR
Makita et al., 2020 [32]	62	95.2%	100.0%	183	68.3%	99.5%	NR	NR	NR	19	68.4%	100%
Rubio et al., 2020 [21]	135	97.8%	98.5%	235	73.2%	71.4%	22	100.0%	100.0%	NR	NR	NR
Hamzah et al., 2020 [33]	20	100.0%	95.0%	56	71.4%	98.2%	NR	NR	NR	NR	NR	NR
Hersi et al., 2021 [34]	163	98.2%	97.5%	351	86.6%	84.9%	33	100.0%	97.0%	NR	NR	NR
Hersi et al., 2021 [34]	165	100.0%	100.0%	371	83.3%	96.8%	29	100.0%	100.0%	195	100.0%	100.0%
Giménez-Climent et al., 2021 [35]	89	96.6%	97.8%	129	90.6%	98.4%	21	85.0%	100.0%	23	86.4%	95.4%
Vidya et al., 2022 [36]	107	92.3%	98.1%	202	96.65%	93.1%	31	100.0%	100.0%	50	100.0%	100.0%

Calculated detection rate (%) of patients, SLNs, positive patients (patients with metastatic sentinel lymph nodes), and positive SLNs (metastatic sentinel lymph nodes) using two methods were listed as percentages. Abbreviations: SM, standard method; SPIO, superparamagnetic iron oxide; SLNs, sentinel lymph nodes; NR, not reported.

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
