# Peer review of "The Application of Magnetic Nanoparticles for Sentinel Lymph Node Detection in Clinically Node-Negative Breast Cancer Patients: A Systemic Review and Meta-Analysis"

_cancers, 2022, doi:10.3390/cancers14205034_

Round 1
Reviewer 1 Report
Major revision
1. If additional analysis is possible by dividing it into dual method (radioactive isotope + blue dye) and single method (radioactive isotope) of SLNB, it is necessary to add it.
Minor revision
1. The reasons for the exclusion of 235 people in box 4 of figure 1 need to be explained.
Author Response
#Reviewer 1
Major revision
- If additional analysis is possible by dividing it into dual method (radioactive isotope + blue dye) and single method (radioactive isotope) of SLNB, it is necessary to add it.
Thanks for your valuable advice. Limitations of this analysis include the variable use of blue dye, which was not standardized across studies. Hence, we underwent the subgroup analysis of the selection of standard method and added the results to the manuscript, you can find it at the manuscript (Line 127-130, Line 249-250, Line273-276, and Figure S1-2).
Minor revisions
- The reasons for the exclusion of 235 people in box 4 of figure 1 need to be explained.
In the process of literature selection, after removing 159 duplicates, screening of the 277 titles and abstracts was performed and a total of 235 articles were excluded. The 235 articles consist of animal studies, conference abstracts, unrelated clinical studies, and so on. We have revised it in the manuscript (Line 143-144).
Reviewer 2 Report
The manuscript is well-written although an language revision would improve its readness.
Although the topic is relevant, there are some reviews published in this field in the past ten years. I would like to know what is the novelty in this manuscript specifically.
It would be good if the authors revise the abbreviations and acronyms.
Between lines 61-77 there are no references and the description is a little confusing. I would suggest to improve the readness of this section and include a couple of updated references.
When I was reading the lines 106-107 I got confused when authors cite "positive patients". Afterwards it became clear that these positive patients were confirmed by biopsy. Please clarify it in the manuscript.
In lines 23-24, line 126 and lines 330-331, and throughout the text, authors discuss about ~lower dose~ of SPIO, but the results are shown as "volume". It would be important to represent the results as actual dosage. We do not know if, in the future, pharmaceutical laboratories will provide variable dosage forms for these products, so it could become confusing for readers. Plus, conceptually, it is wrong to say that 0.5 mL is the dose of a medicine.
In addition to this, authors discuss, in lines 23-24 and line 330-331 that one of the advantages of using SPIO is that this product can be used in lower doses. This affirmation lacks scientific soundness. It is not correct to compare doses for variable active ingredients. Each active ingrediet possess a potency but this parameter, by itself, is not enough to justify the use of one or another, because other parameters will influence, including toxicity. We can compare doses only for the same active ingredients. If that`s what the authors mean, it would be important to specify what`s the usual dose for SPIO and say that the results show that even at doses equivalent to a fraction of the usual dose (that should be calculated based on the usual dose) is enough for getting an adequate result.
At last, authors discuss, in lines 348-352 that the time prior surgery can strongly influence the results, but they do not show these data. It would be important to include a figure and a brief paragraph showing these results.
Author Response
#Review 2
The manuscript is well-written although an language revision would improve its readness.
We apologize for the language problems in the original manuscript. The language presentation was improved with assistance from a native English speaker with appropriate research background. We modified throughout the text according to the comment.
Although the topic is relevant, there are some reviews published in this field in the past ten years. I would like to know what is the novelty in this manuscript specifically.
We are grateful for the suggestion. We searched the literature from the PubMed, Embase, Web of science and Cochrane libraries. To our best of knowledge, there were 3 published meta-analyses closest to our work (Ann Surg Oncol (2016) 23:1508–1514; BJS 2016; 103: 1409–1419 and BJS Open 2019 Vol. 3 Issue 4 Pages 445-452), the latest work published in 2019(BJS Open 2019 Vol. 3 Issue 4 Pages 445-452). After that at least 7 new studies released, and an updated analysis could be necessitated.
The current meta-analysis synthesized data from 19 cohort studies, this work provided enhanced evidence that SPIO provided an alternative to standard method for axillary node detection, and suggests generalizability of the technique to a wider population. The main outcomes of this manuscript are partly similar to the above published reviews. However, the present manuscript has provided more details about the comparison of detection efficiency regard as patients with positive sentinel lymph nodes, and positive sentinel lymph nodes. In addition, a subgroup analysis based on injection doses was conducted in this study for the first time, although this requires further investigation.
We thought our work could attract the researcher’s attention from surgeons and scientists who engaged in the fields of sentinel lymph node tracing in cancer surgery, because we have provided more comprehensive and detailed information about the comparison of SPIO and standard method. If this technique approach could be promoted and propagated in clinical application, patients with carcinoma such as breast cancer, melanoma, oral cavity squamous cell carcinoma, and endometrial cancer, who require SLNB will profit in the long run. For this reason, we submitted the current manuscript.
It would be good if the authors revise the abbreviations and acronyms.
Thank you for underlining this deficiency. Modification has been made throughout the text according to this comment.
Between lines 61-77 there are no references and the description is a little confusing. I would suggest to improve the readness of this section and include a couple of updated references.
We are grateful for the suggestion. To be clearer and in accordance with the reviewer concerns, we have improved the content of this section and added updated references (Line 52-78).
When I was reading the lines 106-107, I got confused when authors cite "positive patients". Afterwards it became clear that these positive patients were confirmed by biopsy. Please clarify it in the manuscript.
We are grateful to reviewer for pointing out this problem. We have modified this expression and attached an additional explanation in the manuscript according to the comment (Line 107-109).
In lines 23-24, line 126 and lines 330-331, and throughout the text, authors discuss about ~lower dose~ of SPIO, but the results are shown as "volume". It would be important to represent the results as actual dosage. We do not know if, in the future, pharmaceutical laboratories will provide variable dosage forms for these products, so it could become confusing for readers. Plus, conceptually, it is wrong to say that 0.5 mL is the dose of a medicine.
Doses related diagnostic efficacy has been discussed in several studies (Ejso 2020,46(12):2195-2201; Cancers 2021, 13, 693). These results provide more evidence that, not only can a smaller dose be equally efficient, but also the cost-reduction and less adverse events could be achieved. Based on such a notion, subgroup analysis based on injection doses have been conducted and presented in our manuscript.
At the time this study was initiated, detailed information of the SPIO used by the incorporated studies, such as concentration and the injection doses, has been carefully collected and evaluated (Listed as table S2). Our investigation found that the concentration of the iron (the core component of the SPIO), even though produced by different manufacturers, is extremely similar (27mg iron per ml, 27.9mg iron per ml, or 28mg iron per ml). In the present analysis, minor differences of the initial concentration were ignored. Ultimately, we used the injection volume of the SPIO to present the actual dosage as described in the previous literature (Ejso 2020,46(12):2195-2201).
Appropriate explanation will be added in the Method part, and this will be discussed in detail as the limitations of our study (Line 127-130, Line 386-387).
In addition to this, authors discuss, in lines 23-24 and line 330-331 that one of the advantages of using SPIO is that this product can be used in lower doses. This affirmation lacks scientific soundness. It is not correct to compare doses for variable active ingredients. Each active ingre diet possess a potency but this parameter, by itself, is not enough to justify the use of one or another, because other parameters will influence, including toxicity. We can compare doses only for the same active ingredients. If that`s what the authors mean, it would be important to specify what`s the usual dose for SPIO and say that the results show that even at doses equivalent to a fraction of the usual dose (that should be calculated based on the usual dose) is enough for getting an adequate result.
Our deepest gratitude goes to you for your rigorous work and specialized comments. By inspecting the information of the used SPIO in the included study, these 3 agents produced by 2 manufactures (Table S2). The core component of the 3 products, which can be detected by a handheld magnetometer, is similar to the initial concentration. Therefore, we used the injection volume of the SPIO to present the actual dosage as described in the previous literature (EJSO 2020,46(12):2195-2201), we pooled the data for subsequent analyses.
Based on incorporated literature, 2ml of SPIO has been the usual injection volume in practice (14 of 19 studies). Lower doses of the SPIO (0.5ml, 1ml or 1.5ml of the injection volume), with similar initial concentration, has also been explored in practice. Actual used doses, converted into the fraction of the usual dose, will be more viable and accurate. Modified throughout the text according to the comment (Line321-332, Figure 3 and Figure 5)
At last, authors discuss, in lines 348-352 that the time prior surgery can strongly influence the results, but they do not show these data. It would be important to include a figure and a brief paragraph showing these results.
We are grateful for the suggestion. To be clearer and in accordance with the reviewer concerns, we have amended the sentence according to the comment (Line 333-334).